# Accuracy of deep learning-based computed tomography diagnostic system for COVID-19: A consecutive sampling external validation cohort study

**Tatsuyoshi Ikenoue**[1☯]*, **Yuki Kataoka**[2,3☯], **Yoshinori Matsuoka**[4], **Junichi Matsumoto**[5], **Junji Kumasawa**[6], **Kentaro Tochitatni**[7], **Hiraku Funakoshi**[8], **Tomohiro Hosoda**[9], **Aiko Kugimiya**[10], **Michinori Shirano**[11], **Fumiko Hamabe**[12], **Sachiyo Iwata**[13], **Shingo Fukuma**[1], **Japan COVID-19 AI team**[¶]

1 Human Health Sciences, Kyoto University Graduate School of Medicine, Kyoto, Japan, 2 Hospital Care Research Unit, Hyogo Prefectural Amagasaki General Medical Center, Amagasaki, Hyogo, Japan, 3 Department of Respiratory Medicine, Hyogo Prefectural Amagasaki General Medical Center, Amagasaki, Hyogo, Japan, 4 Department of Emergency Medicine, Kobe City Medical Center General Hospital, Kobe City, Hyogo, Japan, 5 Department of Emergency and Critical Care Medicine, St. Marianna University School of Medicine, Kawasaki, Kanagawa, Japan, 6 Department of Critical Care Medicine, Sakai City Medical Center, Sakai, Osaka, Japan, 7 Department of Infectious Diseases, Kyoto City Hospital, Kyoto-city, Kyoto, Japan, 8 Department of Emergency and Critical Care Medicine, Tokyobay Urayasu Ichikawa Medical Center, Urayasu, Chiba, Japan, 9 Department of Infectious Disease, Kawasaki Municipal Kawasaki Hospital, Kawasaki-ku, Kawasaki Kanagawa, Japan, 10 Department of Emergency and Critical Care Medicine, Yamanashi Prefectural Central Hospital, Kofu, Yamanashi, Japan, 11 Department of Infectious Diseases, Osaka City General Hospital, Osaka, Japan, 12 Department of Radiology, National Defense Medical College Hospital, Tokorozawa, Saitama, Japan, 13 Division of Cardiovascular Medicine, Hyogo Prefectural Kakogawa Medical Center, Kakogawa, Japan

☯ These authors contributed equally to this work.
¶ Membership of the Japan COVID-19 AI team were listed in the Acknowledgments
* ikenoue.tatsuyoshi.4e@kyoto-u.ac.jp

**Data Availability Statement:** Chest CT images and individual clinical information could not be publicized because of restrictions imposed by the

## Abstract

Ali-M3, an artificial intelligence program, analyzes chest computed tomography (CT) and detects the likelihood of coronavirus disease (COVID-19) based on scores ranging from 0 to 1. However, Ali-M3 has not been externally validated. Our aim was to evaluate the accuracy of Ali-M3 for detecting COVID-19 and discuss its clinical value. We evaluated the external validity of Ali-M3 using sequential Japanese sampling data. In this retrospective cohort study, COVID-19 infection probabilities for 617 symptomatic patients were determined using Ali-M3. In 11 Japanese tertiary care facilities, these patients underwent reverse tran-scription-polymerase chain reaction (RT-PCR) testing. They also underwent chest CT to confirm a diagnosis of COVID-19. Of the 617 patients, 289 (46.8%) were RT-PCR-positive. The area under the curve (AUC) of Ali-M3 for predicting a COVID-19 diagnosis was 0.797 (95% confidence interval: 0.762–0.833) and the goodness-of-fit was $P = 0.156$. With a cut-off probability of a diagnosis of COVID-19 by Ali-M3 set at 0.5, the sensitivity and specificity were 80.6% and 68.3%, respectively. A cut-off of 0.2 yielded a sensitivity and specificity of 89.2% and 43.2%, respectively. Among the 223 patients who required oxygen, the AUC was 0.825. Sensitivity at a cut-off of 0.5% and 0.2% was 88.7% and 97.9%, respectively.

IRB and by Japanese domestic law and guidelines, which do not allow us to open our data according to Article 16 in "Act on the Protection of Personal Information". (http://www.japaneselawtranslation. go.jp/law/detail/?id=2781&vm=04&re=01). The Hyogo Prefectural Amagasaki General Medical Center functioned as the central ethical review committee. Data access requests may be directed to Ms. Kyoko Wasai (contact via agmc.irb@gmail. com).

**Funding:** The authors did not receive financial funding for this study

**Competing interests:** At the start of the study, Ali-m3 existed as a tool produced by Alibaba Damo (Hangzhou) Technology Co., Ltd for research use that had not received any approval. With the results of this study, we have confirmed that the Ali-M3 has clinical benefits. Therefore, under a special expedited review in Japan, Ali-M3 has been approved by the Japanese Pharmaceuticals and Medical Devices Agency (PMDA) and licensed for use as a diagnostic tool in actual practice. For the license in Japan, Ali-m3 should have been a commercial tool. Therefore, it was not planned that M3 Inc would benefit from the commercialization of the Ali-M3 as a result of our research for which M3 provided the Ali-M3 and storage free of charge. The authors have no patents, products in development, or marketed products associated with this research to declare. This does not alter our adherence to PLOS ONE policies on sharing data and materials. In Japan, the M3.inc completely held the right of Ali-M3. This study was utterly free from Alibaba Damo Technology except Alibaba platform by the right of M3.inc.

Although the sensitivity was lower when the days from symptom onset were fewer, the sensitivity increased for both cut-off values after 5 days. We evaluated Ali-M3 using external validation with symptomatic patient data from Japanese tertiary care facilities. As Ali-M3 showed sufficient sensitivity performance, despite a lower specificity performance, Ali-M3 could be useful in excluding a diagnosis of COVID-19.

## Introduction

A proper triage system is critical during the COVID-19 pandemic [1, 2]. An improper triage system may be disadvantageous to patients and lead to a waste of personal protective equipment (PPE). An increase in hospital infections through the admission of infected patients to healthcare facilities could result in the collapse of the medical system. Although reverse transcription-polymerase chain reaction (RT-PCR) tests have been developed, the delay in receiving RT-PCR results could hamper appropriate triage.

Computed tomography (CT) is a fast and useful diagnostic tool. Certain studies have reported characteristic COVID-19 findings on chest CT images [3–8]. The use of chest CT images by radiologists has shown a high diagnostic performance for COVID-19. However, radiologists' interpretations vary greatly. This depends on their familiarization with the interpretation of COVID-19 CT images [9]. Therefore, using CT as a diagnostic tool in general clinical practice is challenging in the current pandemic environment.

Diagnostic support systems using artificial intelligence (AI) have the potential to replace many of the routine detection, characterization, and quantification tasks currently performed by radiologists who use their human cognitive abilities [10]. AI can prevent the diagnostic inconsistencies from inter- and intra-reader diagnoses. In China, where the COVID-19 infection originated, many AI systems have been developed to establish a diagnosis of COVID-19 based on chest CT images [11–15]. One such system, Ali-M3, can detect the likelihood of COVID-19 in a range of 0 to 1. It has excellent COVID-19 detection accuracy. Ali-M3 has an accuracy, sensitivity, and specificity of 99.0%, 98.5%, and 99.2%, respectively. Although Ali-M3 has excellent accuracy, it was developed with a virtual population. This consisted of 3,067 examinations for COVID-19, 1,996 for community-acquired pneumonia, and 1,975 for non-pneumonia. These virtual examinations differed from a general population, therefore its' accuracy could be overestimated [16].

To use Ali-M3 to exclude the diagnosis of COVID-19, its' external validity must be evaluated based on the distribution of disease in a real-world setting. We conducted a retrospective cohort study to evaluate the external validity of Ali-M3. We used the Japanese sequential sampling data of patients who underwent RT-PCR tests as well as chest CT for the diagnosis of COVID-19.

## Materials and methods

### Study design

This retrospective cohort study consisted of 11 Japanese tertiary care facilities that provided treatment for COVID-19 in each region of the country. The institutions from which the medical data were obtained are listed in S1 Table. We collected data from the medical records of each institution between April 15 and May 31, 2020. We partially followed the guidelines of the Transparent Reporting of a Multivariable Prediction Model for Individual Prognosis or

Diagnosis Statement to plan and report this study (S2 Table) [17]. The Institutional Review Board of each facility approved the study. The requirement to obtain written informed consent was waived as it was decided that this was an emergent study with public health implications. The accuracy and reliability of the data were confirmed by PMDA during the approval process of Ali-M3.

## Participants

We included patients who underwent both RT-PCR and chest CT for the diagnosis of COVID-19. The potentially eligible participants were identified on the advice of their physician. The physician confirmed that both an RT-PCR test and a chest CT were obtained when the patient presented with symptoms or was suspected of having COVID-19. Detailed information on the inclusion criteria are shown in S3 Table. We selected patients using consecutive sampling methods between January 1 and April 15, 2020. RT-PCR results were extracted from the medical records of the patients at each facility. The patients were excluded when the time interval between the chest CT and the first RT-PCR assay was greater than 7 days.

All available data in the database was used to maximize the power and generalizability of the results.

## Chest CT protocols

All images were obtained using one of the five types of CT systems with the patient in the supine position. The details of the scanning parameters and systems are listed in S4 Table.

## Image analysis

We used a three-dimensional deep learning framework to detect the COVID-19 infections [16]. The details of this model are included in the S1 File. The population development characteristics from the datasheet are shown in S5 Table. The learning of Ali-M3 was stopped before the evaluation. We set a cut-off point for the model output at 0.5 as this cut-off point was used during the development stage. The investigators who entered the CT image data into Ali-M3 were blinded to the RT-PCR results.

## Reference standard

The diagnosis of COVID-19 was established by an RT-PCR test. This test detects the nucleic acid of severe acute respiratory syndrome coronavirus 2 (SARS-CoV-2) in the sputum, throat swabs, or secretions of lower respiratory tract samples [18]. We established RT-PCR tests as the main reference standard. Although the findings of the chest CT, interpreted by radiologists, were included as the reference standard in the derivation study, we did not include it as the reference standard in the present study.

## Statistical analysis

Statistical analysis was performed using R statistical software, version 3.6.3, (R Foundation for Statistical Computing). Data analysis was performed using a complete case dataset. Continuous variables were presented as means (standard deviation) and categorical variables were presented as counts and percentages. Using the RT-PCR results as a reference, the area under the curve (AUC), sensitivity, specificity, positive-predictive value, and negative-predictive value of the likelihood of COVID-19 (as derived from the Ali-M3's analysis of the chest CT imaging) were calculated. A 95% confidence interval (CI) was determined using the Wilson score

method. The goodness-of-fit was calculated using the Le Cessie–Van Houwelingen normal test statistic for the unweighted sum of squared errors.

## Sensitivity analysis

**Moving cut-off point.**  The objective of this study was to determine whether the AI model could be used as a screening tool for COVID-19 in the real world. In a clinical situation, physicians require an accurate diagnosis of COVID-19. Therefore, they insist on more sensitivity than specificity. For the sensitivity analysis, we moved the cut-off point and observed sensitivities and specificities to minimize the possibility of omitting COVID-19 patients.

**Simulation of imperfect reference.**  In the main analysis, we assumed RT-PCR to be the perfect reference (100% sensitivity and 100% specificity). However, in the real world, RT-PCR is not the perfect reference. Its' sensitivity has been estimated to be 0–80% [19]. To evaluate the effect of this imperfect reference, we calculated the sensitivity, specificity, and AUC of Ali-M3. We used the methods and R code described in the S2 File while varying the sensitivity. However, we established the specificity of RT-PCR at 100% [20].

**Effect of the number of days after symptom onset.**  The number of days that passed before the onset of symptoms affects the presence of antibodies and the performance of RT-PCR tests in COVID-19 patients [19, 21]. However, it is not clear if this could affect CT images in these patients. Sensitivity and specificity were calculated for a group of patients whose symptom onset dates were known. This was calculated for those patients with the elapse of 14 days or more after symptom onset. This was also calculated for patients every 2 days from 0 to 13 days after symptom onset.

**Effect of symptom severity.**  Imaging is not routinely used as a screening test for COVID-19 in asymptomatic individuals [22]. However, CT images were used to assess disease severity. We established the severity by evaluating whether oxygen therapy was required and if the patient was asymptomatic while undergoing CT.

**Effect of reconstruction slice.**  The thickness of the reconstruction slice can affect diagnostic performance [23]. We separated the dataset for the main analysis with a 3-mm thick reconstruction slice. We did this because of the fissure in our data set between 3 mm and 4 mm. We then calculated the performance of the model for each dataset.

## Results

### Study population characteristics

Fig 1 shows the patient flow diagram. Data from 749 patients were analyzed. In this validation study, we assessed 617 symptomatic patients. The characteristics of the study population for the main datasets are listed in Table 1. Overall, 289 patients (46.8%) were diagnosed with COVID-19 using RT-PCR. Thirteen patients required more than two RT-PCR tests before being diagnosed with COVID-19. The major symptoms were dry cough (37.6%), fever (33.5%), and sore throat (25.8%).

### Model performance

The performance of the confidence score after validation among the symptomatic patients is shown in Fig 2. The performance of the confidence score was $P = 0.156$ for the goodness-of-fit and the AUC was 0.797 (95% CI 0.762–0.833). The relationship between the score and the predicted probability is shown in Fig 2. The optimal cut-off point with maximal sensitivity and specificity was 0.5. The sensitivity and specificity were 80.6% (233 of 289), [95% CI: 75.6–85.0%] and 68.3% (224 of 328), [95% CI, 63.3%–93.3%], respectively.

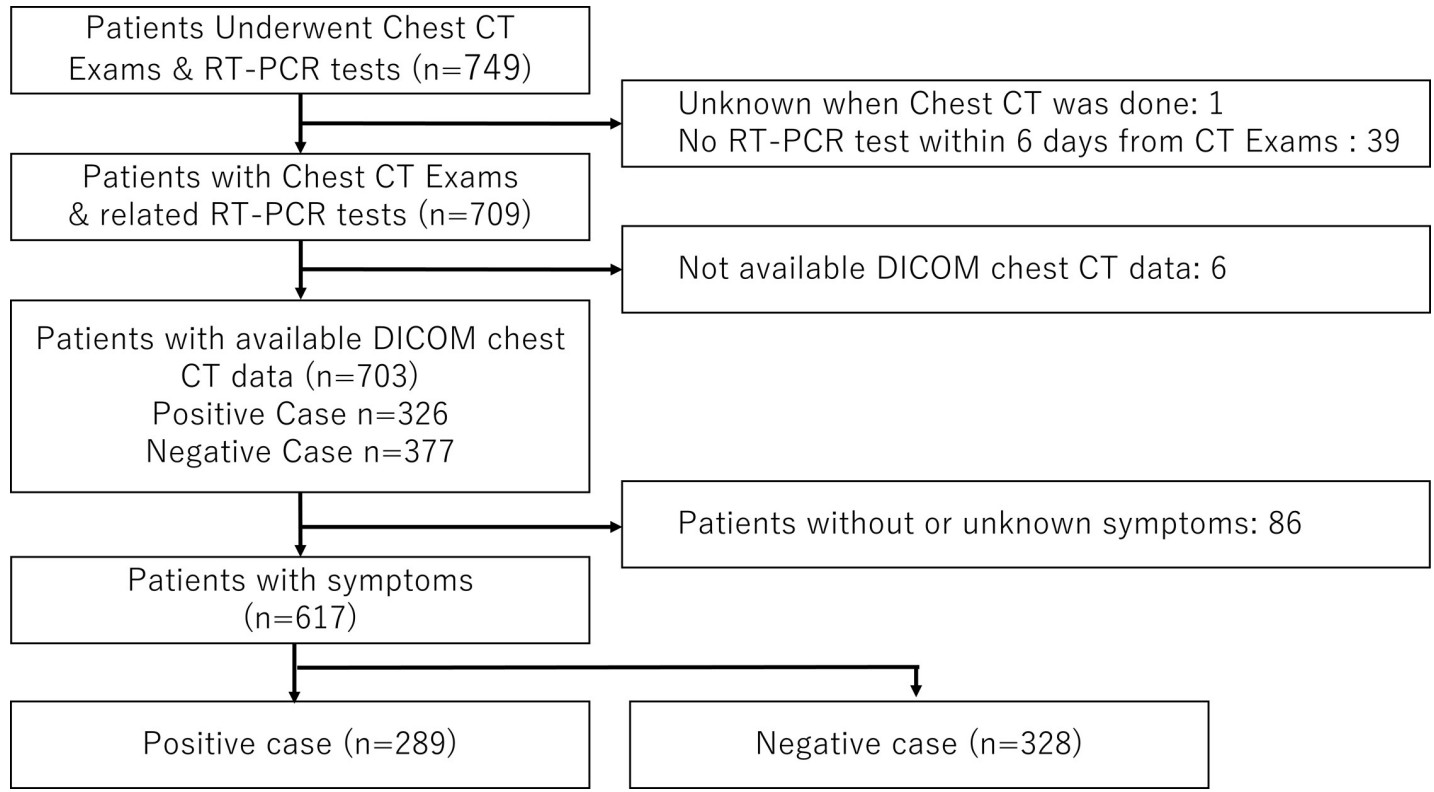

**Fig 1. Patient flow.** Abbreviations: CT, computed tomography; RT-PCR, reverse transcription-polymerase chain reaction; DICOM, digital imaging and communications in medicine.

## Sensitivity analysis

**Moving cut-off point.** Table 2 shows the relationship between the cut-off points for confidence score and performance. When the cut-off point was 0.2, the sensitivity and specificity were 89.2% and 43.3%, respectively.

**Simulation of imperfect reference.** Fig 3 shows the sensitivity and specificity with the assumption of imperfect reference for the RT-PCR test. The AUC was 0.865. When the cut-off point was set at 0.5. using the Youden Index, the sensitivity and specificity were 80.6% and 81.3%, respectively. When the cut-off point was set at 0.2, the sensitivity and specificity were 89.2% and 51.9%, respectively.

**Effect of number of days after symptom onset.** Of all symptomatic patients, 600 (97.2%) were included in the sensitivity analysis. Of these, 17 patients did not know the number of days after symptom onset. Fig 4 shows the relationship between the test performance and the number of days since the onset of symptoms when the confidence score of Ali-M3 was set at 0.5 0.2. Sensitivity values began at 0.7 and increased up to 1.0, until 10–11 days in both cases. However, the specificity values remained similar across the strata. The sensitivity increased over 0.9 when the confidence score was set at 0.2. This was greater than when the confidence score was set at 0.5.

**Changing the eligibility criteria.** The effects of changing the criteria for patient eligibility are shown in Fig 5.

**Dataset focused on asymptomatic patients.** There were 86 asymptomatic patients (RT-PCR positive, n = 37). Using these patients only, the AUC was 0.623. When the cut-off

**Table 1. Demographics of patient characteristics.**

| Variable | Symptomatic patients | Patients using oxygen | Asymptomatic patients |
|---|---|---|---|
| N | 617 | (223) | (86) |
| Age (years old) [+] | 59.6 (19.2) | 68.3 (16.4) | 54.5 (22.4) |
| Sex (Male) | 377 (61.2) | 158 (70.9) | 40 (46.5) |
| Real-time PCR test (Positive) | 289 (46.8) | 97 (43.5) | 37 (43.0) |
| Body temperature ($\geq 37°$) | 391 (66.5) | 143 (69.8) | |
| Systolic Blood Pressure ($\leq 90$ mmHg) | 18 (3.2) | 11 (5.2) | |
| Pulse ($\geq 120$ bpm) | 48 (8.2) | 22 (10.2) | |
| Respiratory rate ($\geq 25$ /minute) | 92 (20.5) | 64 (38.3) | |
| Saturation of percutaneous oxygen ($\leq 92\%$) | 105 (17.7) | 62 (28.7) | |
| Oxygen use | 223 (36.1) | 223 (100.0) | |
| Vasopressor use | 14 (2.3) | 14 (6.3) | |
| Distribution of symptoms reported | | | |
| Dry cough | 232 (37.6) | 67 (30.0) | |
| Chills | 91 (14.7) | 40 (17.9) | |
| Sore throat | 159 (25.8) | 38 (17.0) | |
| Diarrhea | 66 (10.7) | 17 (7.6) | |
| Joint or muscle pain | 46 (7.5) | 12 (5.4) | |
| Conjunctivitis | 30 (4.9) | 9 (4.0) | |
| Loss of smell or taste | 55 (8.9) | 21 (9.4) | |
| Exposure history | | | |
| No | 484 (78.4) | 191 (85.7) | 62 (72.1) |
| Within family | 39 (6.3) | 11 (4.9) | 6 (7.0) |
| Other persons | 94 (15.2) | 21 (9.4) | 18 (20.9) |
| Any international travel | 44 (7.1) | 6 (2.7) | 9 (10.5) |
| Current Smoking | 99 (16.0) | 41 (18.4) | 11 (12.8) |
| Past medical history | | | |
| Cardiac artery disease | 46 (7.5) | 24 (10.8) | 4 (4.7) |
| Stroke | 60 (9.7) | 34 (15.2) | 2 (2.3) |
| Chronic heart failure | 69 (11.2) | 43 (19.3) | 4 (4.7) |
| Chronic kidney disease | 58 (9.4) | 33 (14.8) | 7 (8.1) |
| Chronic obstructive pulmonary disease | 69 (11.2) | 34 (15.2) | 7 (8.1) |
| Malignancy | 105 (17.0) | 62 (27.8) | 8 (9.3) |
| Immune deficiency | 32 (5.2) | 17 (7.6) | 1 (1.2) |
| Hypertension | 119 (19.3) | 71 (31.8) | 11 (12.8) |
| Diabetes | 116 (18.8) | 64 (28.7) | 13 (15.1) |
| Any other disease | 188 (30.5) | 73 (32.7) | 29 (33.7) |

PCR, polymerase chain reaction; bpm, beats per minute

*Patients using oxygen were included in the symptomatic patients.

+ is continuous data, and the others are count data. Continuous variables are expressed as mean (SD) and count data as numbers (percentages).

point was 0.5, the sensitivity and specificity were 51.4% and 59.2%, respectively. When the cut-off point was 0.2, the sensitivity and specificity were 44.9% and 73.0%, respectively.

**Dataset focused on patients requiring oxygen therapy.** A total of 223 patients required oxygen (RT-PCR positive: 97). When using only these patients, the AUC was 0.828. When the cut-off point was set at 0.5, the sensitivity and specificity were 88.7% and 57.9%, respectively. When the cut-off point was set at 0.2, the sensitivity and specificity were 97.9% and 34.9%, respectively.

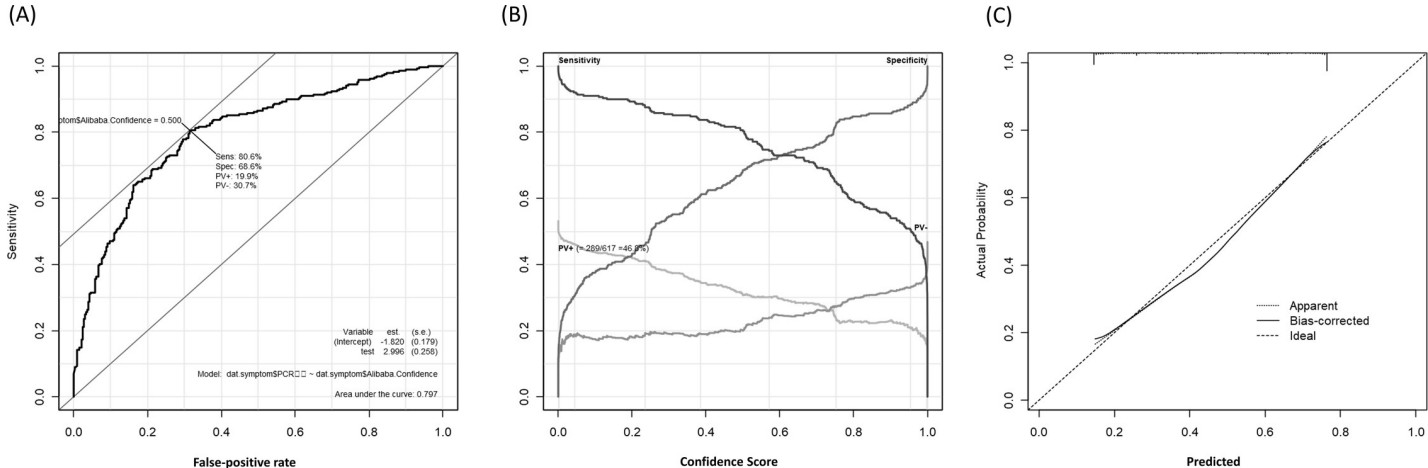

**Fig 2. Differential performance of Ali-M3 for coronavirus disease in symptomatic patients.** (A) A plot of test sensitivity (y-coordinate) versus its' false-positive rate (x-coordinate) obtained at each cutoff level confidence score. The area under the receiver operating characteristic curve is 0.797 and the Youden index is 0.50. (B) A plot of test sensitivity, specificity, positive predictive value (PV+), and negative predictive value (PV-) in y-coordinate versus confidence score obtained from Ali-M3 in x-coordinate. The PV+ is dark gray and the PV- is light gray. The maximum PV+ is 46.8% and the maximum PV- is 53.2%. (C) This graph shows the goodness of fit. The dashed line is an ideal line that predicts the probability obtained from the confidence score of Ali-M3 equal to the actual probability. The pointed line is the fitted line that is estimated with non-linear assumption alone. The dashed line is the fitted line that is estimated with non-linear assumption and considering the bias in nonparametric estimation using the le Cessie-van Houwelingen method.

## Effect of the thickness of the CT reconstruction slice of CT

There were 320 patients (RT-PCR positive: 121) with a reconstruction slice thickness of less than 3-mm. When considering these patients only, the AUC was 0.825. When the cut-off point was set at 0.5, the sensitivity and specificity were 82.6% and 69.7%, respectively. When the cut-off point was set at 0.2, the sensitivity and specificity were 94.2% and 51.5%, respectively. In patients with a reconstruction slice thickness > 3 mm, the AUC was 0.789 (S1 Fig).

## Discussion

In this external validation study, our results indicated that Ali-M3 could be useful for the immediate triage of suspected COVID-19 patients with symptoms at a lower cut-off value. In particular, greater accuracy was observed in patients with greater severity, a few days after symptom onset, and with images with a thinner reconstructed CT slice.

Currently, all patients with symptoms such as fever are triaged as COVID-19 patients. Therefore, medical practitioners must use PPE for each patient [24]. Additionally, bed zoning is essential to avoid contamination of non-infected patients [25]. On the other hand, under-triaging results in hospital infections through the admission of infected patients to health care facilities. This should be continued until a definitive diagnosis is established. Since Ali-M3 is available on the cloud, the physician can receive results immediately. This is accomplished by

**Table 2. Moving cut-off confidence score and test performance.**

| Confidence score | 0.50 | | | | | | | 0.40 | | | | | | | 0.30 | | | | | | | 0.20 | | | | | | | 0.10 | | | | | | |
|---|---|---|---|---|---|---|---|---|---|---|---|---|---|---|---|---|---|---|---|---|---|---|---|---|---|---|---|---|---|---|---|---|---|---|---|
| **Sensitivity** | 0.806 | ( | 0.755 | - | 0.850 | ) | | 0.837 | ( | 0.789 | - | 0.877 | ) | | 0.854 | ( | 0.808 | - | 0.893 | ) | | 0.892 | ( | 0.851 | - | 0.925 | ) | | 0.910 | ( | 0.870 | - | 0.940 | ) | |
| **Specificity** | 0.682 | ( | 0.629 | - | 0.732 | ) | | 0.612 | ( | 0.557 | - | 0.665 | ) | | 0.545 | ( | 0.490 | - | 0.600 | ) | | 0.432 | ( | 0.378 | - | 0.488 | ) | | 0.375 | ( | 0.322 | - | 0.429 | ) | |

AUC (95% confidence interval).

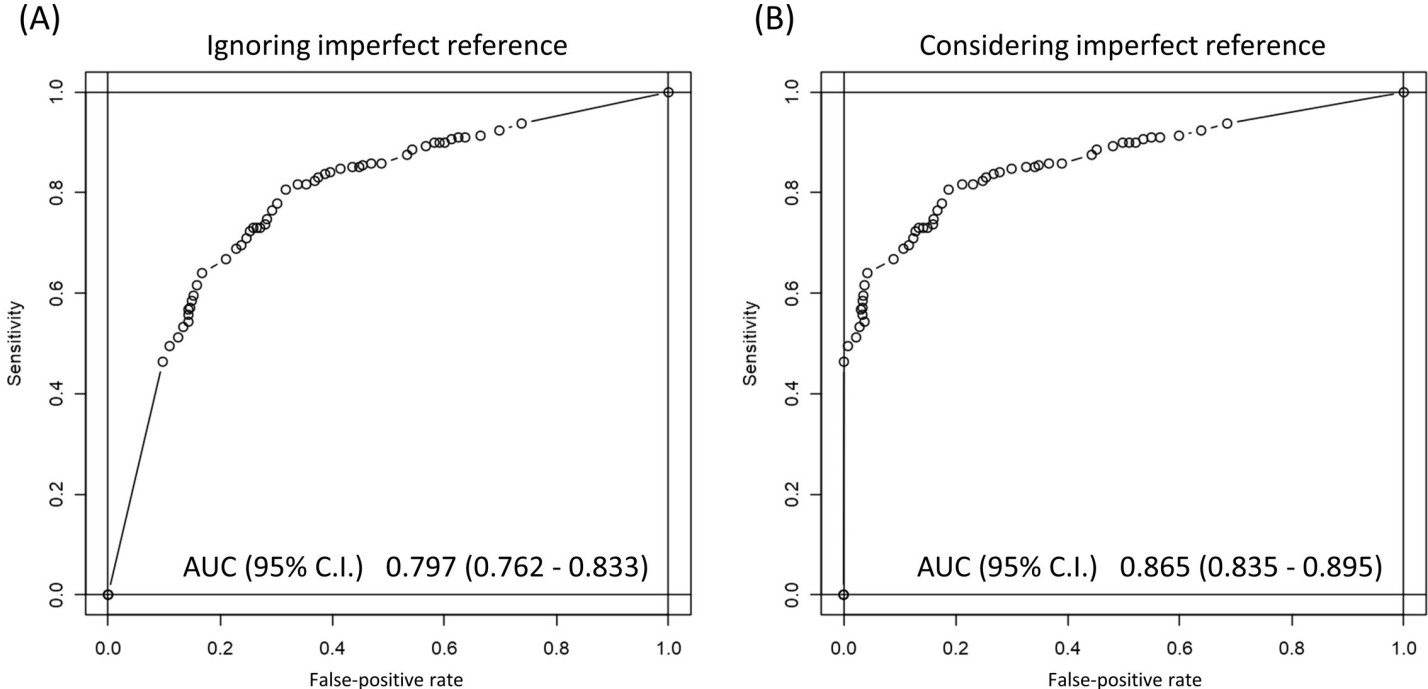

**Fig 3. Relationship between the test performance and the number of days after the onset of symptoms.** (A) The graph shows the relationship between the test performance and the number of days after the onset of symptoms when the confidence score from Ali-M3 is at 0.20. (B) The graph shows the relationship between the test performance and the number of days after onset of symptoms when the confidence score from Ali-M3 is at 0.50. The light gray bar shows the number of patients included in the strata of days after the onset of symptoms, following the right axis. One stratum includes 2 days from day 0 to day 13. The stratum to the extreme right includes 14 days or more. Following the left axis, the solid lines represent the sensitivity in strata, and the dash lines represent specificity in the strata.

sending the digital imaging and communications in the medical images from the ordinal picture archiving and communication system. When applying triage, clinicians require sufficient accuracy in terms of sensitivity. However, the specificity is less important [19]. The high sensitivity obtained at a cut-off of 0.2, with the AI diagnosis, is useful for excluding the diagnosis of COVID-19.

Ali-M3 also has the potential to support the diagnosis of COVID-19. The tools currently used for diagnosing COVID-19 are antibody, antigen, and RT-PCR tests. Both antigen and RT-PCR tests use tracheal secretions or saliva. An antigen test requires an antigen protein above a given detectable level and is currently inferior to the RT-PCR tests. When the same patient sample was used, the antigen test could not support the RT-PCR test. The RT-PCR test is currently used as the gold standard. Although, the sensitivity changes depending on the number of days after the onset of symptoms [19]. Therefore, for an exclusion diagnosis, multiple tests staggered over time are required rather than a single negative RT-PCR test. Even when this test is performed as rapidly as possible, it still requires a few days to obtain multiple test results. On the other hand, Ali-M3 uses the configurational information of the patients' lungs and can add different information. This is apart from that obtained with RT-PCR, thereby complementing the drawbacks of RT-PCR among symptomatic patients with suspected COVID-19.

In this study, the diagnostic accuracy at the validation stage was lower than that at the development stage. A two-gate (case-control) design was used in the development of the AI system. However, in the present study, to evaluate the ability of Ali-M3 to assess a COVID-19 diagnosis by chest CT imaging, we used a single-gate (cohort) design. Although many studies have

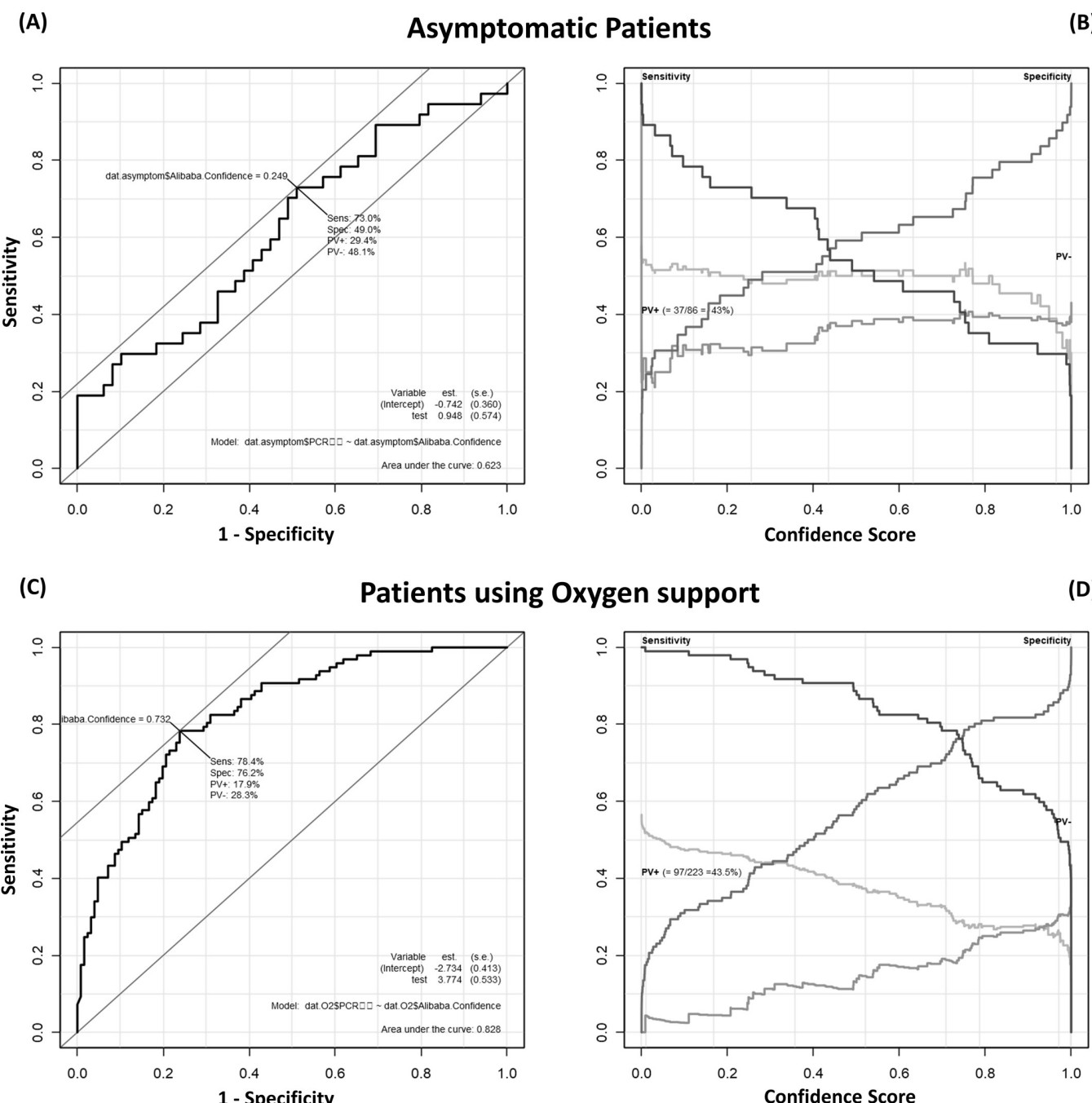

**Fig 4. Receiver operating characteristic (ROC) curves when ignoring imperfect reference and considering imperfect reference.** (A) A plot of test sensitivity (y-coordinate) versus its false-positive rate (x-coordinate) obtained at each cut-off level of confidence score ignoring imperfect reference. The area under the ROC curve is 0.797. (B) A plot of test sensitivity (y-coordinate) versus its false-positive rate (x-coordinate) was obtained at each cut-off level confidence score considering imperfect reference. The area under the ROC curve is 0.865.

used the two-gate design for the evaluation of AI for the diagnosis of COVID-19 [26], the two-gate design is generally prone to overestimation of diagnostic test results [27]. Thus, blindly using the results of a two-gate design in a clinical situation can be inappropriate. Moreover, other factors must be considered. With the use of a two-gate design, the fact that RT-PCR is an

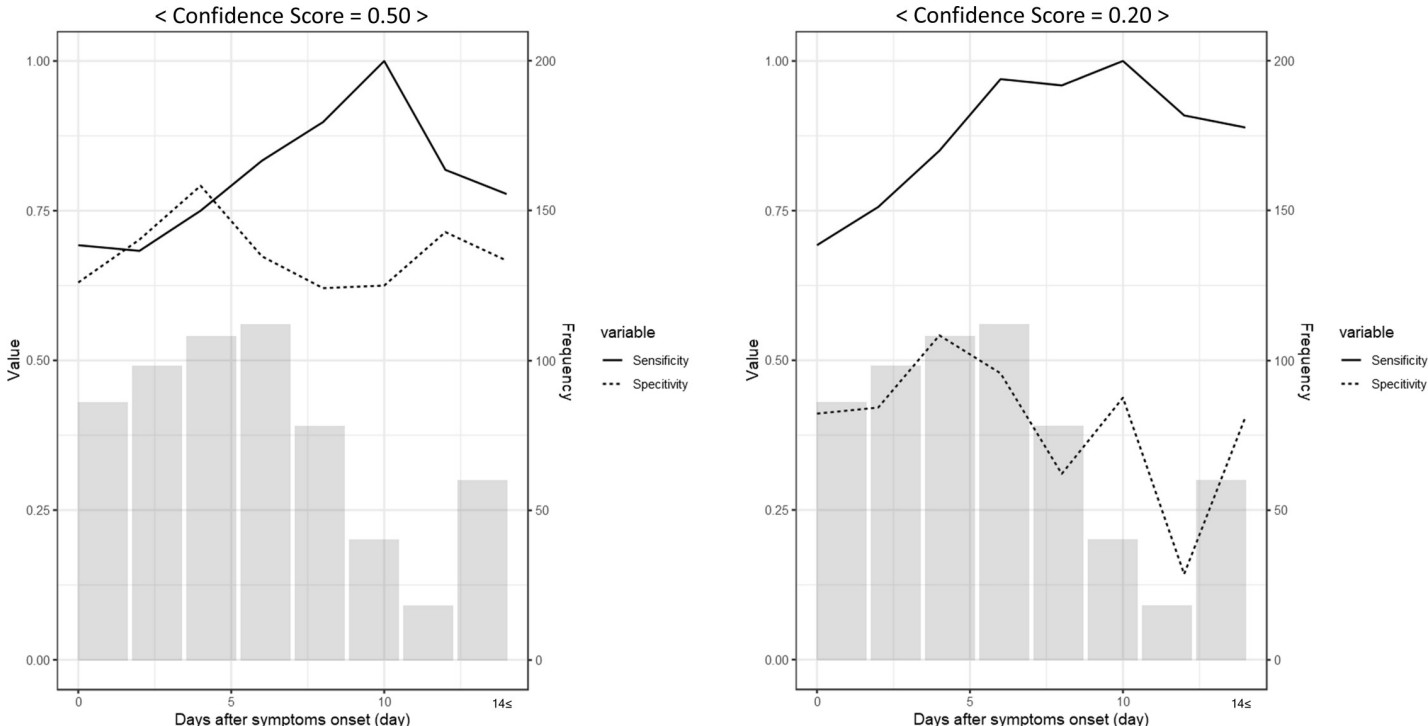

**Fig 5. Differential performance of Ali-M3 for Covid-19 in asymptomatic patients and patients using oxygen.** (A) A plot of test sensitivity (y-coordinate) versus its' false-positive rate (x-coordinate) obtained at each cut off level confidence score in asymptomatic patients. The area under the receiver operating characteristic (ROC) curve is 0.623 and the Youden index is 0.25. (B) A plot of test sensitivity, specificity, positive predictive value (PV+), and negative predictive value (PV-) in y-coordinate versus the confidence score obtained from Ali-M3 in x coordinate among asymptomatic patients. The PV+ is dark gray and PV- is light gray. The maximum PV+ is 43.0% and maximum PV- is 57.0%. (C) A plot of test sensitivity (y-coordinate) versus its' false-positive rate (x-coordinate) obtained at each cut off confidence score level in patients using oxygen. The area under the ROC curve is 0.623 and the Youden index is 0.25. (D) A plot of test sensitivity, specificity, PV+, and PV- in y-coordinate versus confidence scores obtained from Ali-M3 in x-coordinate in patients using oxygen. The PV+ is dark gray and the PV- is light gray. The maximum PV+ is 43.5% and the maximum PV- is 56.5%.

imperfect reference standard is typically ignored. Furthermore, performing culture and tests to ascertain the true sensitivity of this test is difficult. In the present study, we simulated the diagnostic ability of Ali-M3 considering that the sensitivity of the reference standard was imperfect. This leads to an underestimation of the specificity and AUC of Ali-M3, without distortion of the sensitivity. Furthermore, the outcomes of developing Ali-M3 and examining its' adequacy were different. Taking into account the patient flow in China, the outcomes at the development stage were set as positive cases with negative RT-PCR results and positive CT image findings [28]. This had a small effect on the sensitivity, but a large effect on the specificity. For example, in the development stage, 33.9% of the positive patients had negative RT-PCR results and positive CT image findings [28]. The performance showed a sensitivity of 98.5% and a specificity of 99.2% during the development of Ali-M3 [16]. A change from 97.7% to 100% for sensitivity and from 80.8% to 81.6% for specificity takes place when a positive RT-PCR result is the only reference. Upgrading to a diagnostic AI that targets only RT-PCR-positive cases at the developmental stage is desirable.

This study had some limitations. First, the differentiation performance of Ali-M3 was poor in asymptomatic patients and Ali-M3 did not show good specificity even if the cut-off was changed. Thus, Ali-M3 should not be used to screen asymptomatic patients. While an alternative to the RT-PCR test for COVID-19 is expected in terms of screening for nosocomial infections and screening on admission for patients with other diseases, Ali-M3 is not

recommended for this purpose. Second, we could not differentiate COVID-19 from other forms of viral pneumonia. Compared to the past five seasons, the number of Japanese people infected with influenza during this season was markedly low [29]. Only a few cases in our cohort were diagnosed with other forms of viral pneumonia. Third, Ali-M3 could not reflect the differences in imaging features caused by different COVID-19 types. In addition to type A COVID-19, which was initially prevalent in Asia, type B and type C were prevalent in Europe and the United States. These different types were not determined in the PCR test. Thus, we could not evaluate these differences. Fourth, the AI system, generally known as the decision process, is a black-box system. Although Ali-M3 also has the aspects of a black-box, it shows imagines that are the cause of the decision. [16].

## Conclusions

We conducted a retrospective cohort study for the external validation of Ali-M3 using symptomatic patient data from Japanese tertiary care facilities. Despite limited data analysis, our results indicated that AI-based CT diagnosis could be useful for a diagnosis of the exclusion of COVID-19 in symptomatic patients. This is particularly true in patients requiring oxygen and only a few days after symptom onset. Using Ali-M3 support can reduce PPE consumption and prevent hospital infections through the admission of covertly infected patients. Moreover, Ali-M3 also has the potential to support the diagnosis of RT-PCR in patients with suspected COVID-19. However, as Ali-M3 has some limitations in terms of development, further studies and learning are warranted to update this system.

## Supporting information

**S1 Fig. Differential performance of Ali-M3 for coronavirus disease in patients divided by the thickness of the reconstructed slice of computed tomography.** (A) A plot of test sensitivity (y coordinate) versus its' false-positive rate (x coordinate) obtained at each cutoff level confidence score under the 3 mm thickness of the reconstruction slice. The area under the receiver operating characteristic (ROC) curve was 0.825 and the Youden index was 0.50. (B) A plot of test sensitivity, specificity, positive predictive value (PV+), and negative predictive value (PV-) in the y coordinate versus the confidence score obtained from Ali-M3 in the x coordinate under the 3 mm thickness of the reconstruction slice. PV+ is dark gray, and PV- is light gray. The maximum PV+ was 46.5%, and the maximum PV- was 53.5%. (C) A plot of test sensitivity (y coordinate) versus its' false-positive rate (x coordinate) obtained at each cut-off confidence score level over the 3 mm thickness of the reconstruction slice. The area under the ROC curve was 0.789, and the Youden index was 0.50. (D) A plot of test sensitivity, specificity, PV+, and PV- in the y coordinate versus the confidence score obtained from Ali-M3 in the x coordinate over the 3 mm thickness of the reconstruction slice. PV+ is dark gray, and PV- is light gray. The maximum PV+ was 47.0%, and the maximum PV- was 53.0%.
(PNG)

**S1 Table. The list of institutions from which patient medical data was obtained.**
(DOCX)

**S2 Table. Checklist of the guidelines of the Transparent Reporting of a Multivariable Prediction.** Model for Individual Prognosis or Diagnosis Statement.
(DOCX)

**S3 Table. Inclusion criteria.** Patients who met the following criteria even for one item were considered symptomatic and were enrolled in the study.
(DOCX)

**S4 Table Computed tomography system and protocol.**
(DOCX)

**S5 Table. Population characteristics in development of Ali-M3 from the datasheet.**
(DOCX)

**S1 File. The datasheet of Ali-M3.**
(PDF)

**S2 File. R code to evaluate the effect of the imperfect reference.**
(DOCX)

## Acknowledgments

We thank M3 Inc. and Clinical Porter for providing free Ali-M3 and data storage, although they did not participate in the preparation protocol and manuscript. The analysis of the CT by Ali-M3 was carried out by Nobori on behalf of M3. (M3 and Nobori did not know the patients' data including the result of RT-PCR) Ali-M3 was officially approved by the Japanese PMDA using our data on June 29, 2020. (Approval number form PMDA: 30200BZX00212000, https://www.pmda.go.jp/english/about-pmda/0002.html) To access the Ali-M3 system please contact M3 (m3-ai-lab@m3.com). We also thank Ms. Kyoko Wasai, who assisted in retrieving data and Editage (http://www.editage.com) for editing and reviewing this manuscript for English language. The group author affiliations were as follows: Shingo Hamaguchi, Takafumi Haraguchi (St. Marianna University School of Medicine), Shungo Yamamoto (Kyoto City Hospital), Hiromitsu Sumikawa, Koji Nishida (Sakai City Medical Center), Haruka Nishida, Koichi Ariyoshi (Kobe City Medical Center General Hospital), Hiroshi Shinmoto, Hiroaki Sugiura (National Defense Medical College Hospital), Hidenori Nakagawa, Tomohiro Asaoka (Osaka City General Hospital), Naofumi Yoshida(Kobe University Graduate School of Medicine), Rentaro Oda (Tokyobay Urayasu Ichikawa Medical Center), Takashi Koyama, Yui Iwai (Hyogo Prefectural Amagasaki General Medical Center), and Yoshihiro Miyashita (Yamanashi Prefectural Central Hospital). the lead author for this group was Koichi Ariyoshi (kobe9914@yahoo.co.jp).

## Author Contributions

**Conceptualization:** Tatsuyoshi Ikenoue, Yuki Kataoka, Yoshinori Matsuoka, Junichi Matsumoto, Junji Kumasawa, Shingo Fukuma.

**Data curation:** Tatsuyoshi Ikenoue, Yuki Kataoka, Yoshinori Matsuoka, Junichi Matsumoto, Junji Kumasawa, Kentaro Tochitatni, Hiraku Funakoshi, Tomohiro Hosoda, Aiko Kugimiya, Michinori Shirano, Fumiko Hamabe, Sachiyo Iwata.

**Formal analysis:** Tatsuyoshi Ikenoue, Yuki Kataoka.

**Funding acquisition:** Tatsuyoshi Ikenoue, Yuki Kataoka, Shingo Fukuma.

**Investigation:** Tatsuyoshi Ikenoue, Yuki Kataoka, Junichi Matsumoto, Junji Kumasawa, Kentaro Tochitatni, Hiraku Funakoshi, Tomohiro Hosoda, Aiko Kugimiya, Michinori Shirano, Fumiko Hamabe, Sachiyo Iwata.

**Methodology:** Tatsuyoshi Ikenoue, Yuki Kataoka.

**Project administration:** Tatsuyoshi Ikenoue, Yuki Kataoka, Yoshinori Matsuoka, Hiraku Funakoshi, Shingo Fukuma.

**Resources:** Tatsuyoshi Ikenoue, Yuki Kataoka, Yoshinori Matsuoka, Junichi Matsumoto, Junji Kumasawa, Kentaro Tochitatni, Hiraku Funakoshi, Tomohiro Hosoda, Aiko Kugimiya, Michinori Shirano, Fumiko Hamabe, Sachiyo Iwata.

**Software:** Tatsuyoshi Ikenoue, Yuki Kataoka.

**Supervision:** Tatsuyoshi Ikenoue, Yuki Kataoka, Yoshinori Matsuoka, Junichi Matsumoto, Junji Kumasawa, Kentaro Tochitatni, Hiraku Funakoshi, Tomohiro Hosoda, Aiko Kugimiya, Michinori Shirano, Fumiko Hamabe, Sachiyo Iwata, Shingo Fukuma.

**Validation:** Tatsuyoshi Ikenoue, Yuki Kataoka, Yoshinori Matsuoka, Junichi Matsumoto, Junji Kumasawa, Kentaro Tochitatni, Hiraku Funakoshi, Tomohiro Hosoda, Aiko Kugimiya, Michinori Shirano, Fumiko Hamabe, Sachiyo Iwata.

**Visualization:** Tatsuyoshi Ikenoue, Yuki Kataoka.

**Writing – original draft:** Tatsuyoshi Ikenoue, Yuki Kataoka.

**Writing – review & editing:** Tatsuyoshi Ikenoue, Yuki Kataoka, Yoshinori Matsuoka, Junichi Matsumoto, Junji Kumasawa, Kentaro Tochitatni, Hiraku Funakoshi, Tomohiro Hosoda, Aiko Kugimiya, Michinori Shirano, Fumiko Hamabe, Sachiyo Iwata, Shingo Fukuma.

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
