## [Decision Letter · Decision Letter 0]

29 Mar 2021

PONE-D-21-03621

Accuracy of deep learning-based computed tomography diagnostic system of COVID-19: A consecutive sampling external validation cohort study

PLOS ONE

Dear Dr. Ikenoue,

Thank you for submitting your manuscript to PLOS ONE. After careful consideration, we feel that it has merit but does not fully meet PLOS ONE’s publication criteria as it currently stands. Therefore, we invite you to submit a revised version of the manuscript that addresses the points raised during the review process.

We look forward to receiving your revised manuscript.

Kind regards,

Haoran Xie

Academic Editor

PLOS ONE

Journal Requirements:

3. Thank you for providing the date(s) when patient medical information was initially recorded (between January 1 and April 15, 2020). Please also include the date(s) on which your research team accessed the databases/records to obtain the retrospective data used in your study.

4. In your methods section or in the supplementary material, please provide the names of the 11 institutions where patient medical data was obtained from.

5. In your methods section, please provide the names and catalog numbers of the RT-PCR tests used in this study.

6. Thank you for stating the following financial disclosure:

"NO"

7. Thank you for stating the following in your Competing Interests section: 

"NO"

8. In your Data Availability statement, you have not specified where the minimal data set underlying the results described in your manuscript can be found. PLOS defines a study's minimal data set as the underlying data used to reach the conclusions drawn in the manuscript and any additional data required to replicate the reported study findings in their entirety. All PLOS journals require that the minimal data set be made fully available. For more information about our data policy, please see http://journals.plos.org/plosone/s/data-availability.

9. One of the noted authors is a group or consortium [Japan COVID-19 AI team]. In addition to naming the author group, please list the individual authors and affiliations within this group in the acknowledgments section of your manuscript. Please also indicate clearly a lead author for this group along with a contact email address.

10. Please include captions for your Supporting Information files at the end of your manuscript, and update any in-text citations to match accordingly. Please see our Supporting Information guidelines for more information: http://journals.plos.org/plosone/s/supporting-information.

Reviewers' comments:

Reviewer's Responses to Questions

**Comments to the Author**

1. Is the manuscript technically sound, and do the data support the conclusions?

Reviewer #1: Partly

Reviewer #2: Yes

Reviewer #3: No

2. Has the statistical analysis been performed appropriately and rigorously? 

Reviewer #1: No

Reviewer #2: Yes

Reviewer #3: No

3. Have the authors made all data underlying the findings in their manuscript fully available?

Reviewer #1: No

Reviewer #2: Yes

Reviewer #3: No

4. Is the manuscript presented in an intelligible fashion and written in standard English?

Reviewer #1: Yes

Reviewer #2: Yes

Reviewer #3: Yes

5. Review Comments to the Author

Reviewer #1: This study carried out an external validation of a commercial tool Ali-m3. This is necessary for the area of AI-based medical systems. A number of concerns should be resolved before a further decision could be made.

1. The tool is a commercial tool on the cloud system, which means the commercial provider may change the code and models as they want. And the source code of Ali-m3 is not publicly available. Please clarify how this study ensure the replicability of this tool Ali-m3.

2. The authors mentioned that their data are unavailable to the public, either. The validation data are simply chest CT images, which are very easy to be anonymized. There are many freely available databases of chest CT images. So the chest CT images, the clinical data, and the diagnosis results of the samples need to be released to the public, after being anonymized. The prediction results of the tool Ali-m3 should also be released to the public for the replication purpose.

3. The free access to the commercial tool and online data storage IS a financial support. Please clarify this in the conflict of interest statement.

4. The current cohort consists of 617 patients, with 289 COVID-19 positive patients, and 223 patients with severe symptoms (needing oxygen support). The practical situation has many more COVID-19 negative patients. Considering the specificity is only 43.2% using the Ali-m3 score threshold 0.2, please clarify how to handle the increasing high number of false positives.

5. The results should be strictly discussed. For example, in the Abstract, “sensitivity increased for both cut-off values after 5 days”. But only one threshold 0.2 was mentioned in the Abstract.

6. And for the “223 patients who required oxygen support”, it’s misleading to skip mentioning the specificity. If we set the threshold to the extreme value (like 0), we can get 100% in sensitivity. But that is not an intelligent tool.

7. The commercial provider for Ali-m3 has a website in Japanese only. It’s impossible to review whether this company is a solid AI company or maybe just a contractor of this tool Ali-m3. So the quality and stability of Ali-m3 is unpredictable.

8. Does Ali-m3 have a medical license approved by some governmental agencies?

9. This study cited the commercial tool Ali-m3 by an internal report of a commercial company, which is not the service provider “m3”. Please clarify this.

10. And what is the online like to the validated tool Ali-m3? It’s not acceptable to ask the anonymous reviewer to contact the commercial provider to access the cloud-based tool.

Reviewer #2: The manuscript is about a system for real-time sentiment prediction on Twitter streaming data for coronavirus pandemic. The paper is well-organised, but I still have some concerns:

1) In my idea, the paper contributions are not significant. There is no novelty.

2) There is some repetitive information in different parts of the manuscript about Twitter and sentiment analysis, etc.

3) The result part is the written form of tables.

4) The discussion part didn't discuss anything; it's just repeating the result section in other words.

5) There is some punctuation mistake in the manuscript.

Reviewer #3: The contribution of this research paper isn't clear. Sorry to say that, however, I can't get the point of this paper from the manuscript. Although you state your purpose as "Ali-M3, ... However, Ali-M3 has not been externally validated.", this statement didn't show anything about what you want to do in this research paper.

Based on the conclusion of this paper, "Our results indicated that AI-based CT diagnosis could be useful for ...", it seems that you want to prove that Ali-M3 can be used to diagnose COVID-19, but the data samples used to evaluate Ali-M3 and the results are not good enough to support your conclusion. There are only several hundreds of samples in your evaluation process, even more, you didn't provide background information about those samples, such as how were they collected and which groups of people they covered. So, in my opinion, they can't represent all COVID-19 situation.

Besides the insufficient testing samples, the performance of the model with AUC 0.79, 0.82 isn't very good. How could a model with such performance be used in COVID-19 diagnosis?

Another question, what is your work in this research? From the manuscript, I see that you ran the Ali-M3 model which is already a usable deep learning model, with patients data which I don't know you collected it or not, and take some simple analysis about the results. Are these all you had did in this research? What's the significance of what you did? Maybe you could add more contents in your manuscript about what you did, such as data collection, sample pre-processing, model adjustment, deep analysis, diagnosis direction, practice guideline, or some other things.

A lot of analysis were done focusing on cut-off point adjustment. However what's the meaning of those analysis? Sensitivity and specificity have big changes when you use different cut-off values and they can be affected by the ratio of positive and negative samples of testing dataset. So I think it's not necessary to analysis those values because they can't represent real performance of prediction model.

6. PLOS authors have the option to publish the peer review history of their article (what does this mean?). If published, this will include your full peer review and any attached files.

Reviewer #1: No

Reviewer #2: No

Reviewer #3: No

---

## [Author Response · Author response to Decision Letter 0]

1 Sep 2021

Dear Editor,

We appreciate the opportunity to revise our manuscript. Please find below our responses to the editorial comments and reviewers' comments.

Response: Thank you for your comments. We have modified manuscript format.

Response: Thank you for your comments. We have added the title page to the manuscript.

3. Thank you for providing the date(s) when patient medical information was initially recorded (between January 1 and April 15, 2020). Please also include the date(s) on which your research team accessed the databases/records to obtain the retrospective data used in your study.

Response: Thank you for your advice. We added the following sentence to the Materials and Methods section:

Materials and Methods: Study design

We collected data from medical records between April 15 and May 31, 2020.

4. In your methods section or in the supplementary material, please provide the names of the 11 institutions where patient medical data was obtained from.

Response: Thank you for your advice regarding the information from the institutions. We have added the following sentence to the Materials and Methods section and the supplemental material.

Materials and Methods: Study design

The institutions where patient medical data were obtained are listed in Supplemental Table 1.

5. In your methods section, please provide the names and catalog numbers of the RT-PCR tests used in this study.

Response: Unfortunately, no restrictions were placed on RT-PCR testing in this study. For this reason, we were not able to provide the names and catalog numbers of the RT-PCR tests. However, in legitimate medical institutions in Japan, PCR tests were performed using the following methods.

https://www.niid.go.jp/niid/images/lab-manual/2019-nCoV20200319.pdf

Shirato K et al. (2020). Development of genetic diagnostic methods for novel coronavirus 2019 (nCoV-2019) in Japan. Jpn J Infect Dis. 2020 Volume 73 Issue 4 Pages 304-307. DOI: 10.7883/yoken.JJID.2020.061.

Matsuyama S, et al. (2020). Enhanced isolation of SARS-CoV-2 by TMPRSS2-expressing cells. Proc Natl Acad Sci U S A. 2020 Mar 31;117(13):7001-7003. 

Shirato K et al., (2020). Performance evaluation of real-time RT-PCR assays for detection of severe acute respiratory syndrome coronavirus-2 developed by the National　 Institute of Infectious Diseases, Japan. Jpn J Infect Dis. 2021.　 https://www.niid.go.jp/niid/images/jjid/COVID19/No27_2020-1079R1_20210202.pdf.

6. Thank you for stating the following financial disclosure:

"NO"

a. Please clarify the sources of funding (financial or material support) for your study. List the grants or organizations that supported your study, including funding received from your institution.

d. If you did not receive any funding for this study, please state: “The authors received no specific funding for this work.”

Response: Thank you for providing detailed information on the financial disclosure. We have answered your questions individually.

a. No financial funding was received for this study. M3 Inc. and Clinical Porter provided Ali-M3 and data storage for free.

b. M3 Inc. and Clinical Porter did not participate in the preparation protocol or the development of the manuscript. At the initiations of the study, Ali-M3 existed as a tool produced by Alibaba Damo (Hangzhou) Technology Co., Ltd for research use that had not received any approval. It was not known whether it would be of any value. Therefore, the Ali-M3 was not allowed by Japanese law to be used as a diagnostic tool in actual practice. M3 was only one of the channels for Japanese researchers to connect Japanese researchers and Alibaba. We approached M3 to validate Ali-M3 because the lack of validity prevented chest CT use in the diagnosis of COVID-19. However, with the results of this study, we confirmed that Ali-M3 has clinical benefits. Therefore, under a special expedited review in Japan, Ali-M3 has been approved by the Japanese Pharmaceuticals and Medical Devices Agency (PMDA) and licensed for use as a diagnostic tool in actual practice. For the license, Ali-M3 should have been a commercial tool. However, commercializing was against our wishes. We wanted Ali-M3 to be used for free, even if it was for a specific period of time.

c. No author received a salary from M3 Inc. and Clinical Porter.

d. We added the sentence below to the COI section.

# Competing Interests

The authors did not receive any financial funding for this work.

7. Thank you for stating the following in your Competing Interests section: 

"NO"

Response: Thank you for your comment on COI. The authors have no competing interests to declare. We have added the following information to the COI section in the manuscript as follows:

Competing interests (COI).

The authors declare no competing interests. The authors did not receive financial funding for this study. At the initiation of the study, Ali-M3 existed as a tool produced by Alibaba Damo (Hangzhou) Technology Co., Ltd for research use that had not received any approval. The results of this study confirm that Ali-M3 has clinical benefits. Therefore, under a special expedited review in Japan, Ali-M3 has been approved by the Japanese Pharmaceuticals and Medical Devices Agency (PMDA) and licensed for use as a diagnostic tool in actual practice. (Approval number: 30200BZX00212000, Datasheet: https://www.pmda.go.jp/files/000235943.pdf) For the license in Japan, Ali-M3 should have been a commercial tool. Therefore, it was not anticipated that M3 Inc. would benefit from the commercialization of the Ali-M3 because of our research. M3 provided the Ali-M3 and storage free of charge. 

8. In your Data Availability statement, you have not specified where the minimal data set underlying the results described in your manuscript can be found. PLOS defines a study's minimal data set as the underlying data used to reach the conclusions drawn in the manuscript and any additional data required to replicate the reported study findings in their entirety. All PLOS journals require that the minimal data set be made fully available. For more information about our data policy, please see http://journals.plos.org/plosone/s/data-availability.

Response: We regret that chest CT images and individual clinical information could not be publicized. This is because of the IRB's decision and the Japanese guidelines for protecting personal information. This study was performed in a worldwide emergency setting. We received immediate approval from IRB. This was not the routine method of approval. The IRB of each facility approved the study and the need to obtain written informed consent was waived. Japanese guidelines concerning personal information does not allow personal data to be used by third parties without patient consent. On the other hand, the accuracy and reliability of the data were confirmed by PMDA. This was accomplished during the approval process by the comparing raw data in each hospital and analysis data. We modified the methods section for IRB and data to describe specialty and data reliability as follows. 

Materials and Methods: Study design

The Institutional Review Board of each facility approved the study. The requirement to obtain written informed consent was waived as it was decided that this was an emergent study with public health implications. The accuracy and reliability of the data were confirmed by PMDA during the approval process of Ali-M3.

9. One of the noted authors is a group or consortium [Japan COVID-19 AI team]. In addition to naming the author group, please list the individual authors and affiliations within this group in the acknowledgments section of your manuscript. Please also indicate clearly a lead author for this group along with a contact email address.

Response: Thank you for your advice concerning the group authors. We listed the individual authors and their affiliations within the group in the acknowledgments section. The lead author of this group was Koichi Ariyoshi (kobe9914@yahoo.co.jp). We have added this information to the acknowledgments.

10. Please include captions for your Supporting Information files at the end of your manuscript, and update any in-text citations to match accordingly. Please see our Supporting Information guidelines for more information: http://journals.plos.org/plosone/s/supporting-information.

Response: Thank you for your advice regarding Supporting Information files. We have modified the Supporting Information files and citations in the manuscript.

 

Reviewer #1: 

This study carried out an external validation of a commercial tool Ali-m3. This is necessary for the area of AI-based medical systems. A number of concerns should be resolved before a further decision could be made.

Reply:

Thank you for your comments. 

Ali-M3 is a diagnostic tool used for COVID-19 identification on chest CT. Although many Japanese hospitals have CT systems in their facilities, Japanese practitioners cannot effectively use chest CT in the diagnosis of COVID-19. 

At the initiation of the study, Ali-M3 existed as a tool produced by Alibaba Damo (Hangzhou) Technology Co., Ltd for research use that had not received approval. It is not known whether it would be of any value. Therefore, the Ali-M3 was not allowed by Japanese law to be used as a diagnostic tool in actual practice. M3 was only one of the channels for Japanese researchers to connect Japanese researchers and Alibaba. We approached M3 to validate Ali-M3 because the lack of validity prevented chest CT use for the diagnosis of COVID-19. However, with the results of this study, we confirmed that Ali-M3 has clinical benefits. Therefore, under a special expedited review in Japan, Ali-M3 has been approved by the Japanese Pharmaceuticals and Medical Devices Agency (PMDA) and licensed for use as a diagnostic tool in actual practice. For the license, Ali-M3 should have been a commercial tool. However, commercializing was against our wishes. We wanted Ali-M3 to be used for free, even if it was for a specific period of time.

1. The tool is a commercial tool on the cloud system, which means the commercial provider may change the code and models as they want. And the source code of Ali-m3 is not publicly available. Please clarify how this study ensure the replicability of this tool Ali-m3.

Reply:

Thank you for your comment regarding replicability. 

Ali-M3 was a fixed model, and the same model can be obtained on a commercial basis, as we used in this study. To date, no medical device that continues learning after approval has been approved in Japan. For use in academic research, Alibaba Damo (Hangzhou) Technology Co., Ltd provided M3 with a program in which the learning process had already been halted.

The publicity of the source code of Ali-M3 is unavailable because Ali-M3 has already been approved as a diagnostic tool in actual practice and became commercial in this process.

2. The authors mentioned that their data are unavailable to the public, either. The validation data are simply chest CT images, which are very easy to be anonymized. There are many freely available databases of chest CT images. So the chest CT images, the clinical data, and the diagnosis results of the samples need to be released to the public, after being anonymized. The prediction results of the tool Ali-m3 should also be released to the public for the replication purpose.

Reply:

Thank you for your comment regarding external validation using other available data. 

We searched for external data that was a sequential sampling dataset as thoroughly as possible. However, we did not discover any such dataset. We discussed external validation using two-gate designs or single-gate designs in the 3rd paragraph of the Discussion section. The purpose of our study was to perform external validation using a single-gate design. Although many studies have used the two-gate design for the evaluation of AI for the diagnosis of COVID-19, the two-gate design is generally prone to an overestimation of diagnostic test results. Thus, blindly using the results based on a two-gate design in a clinical situation can be inappropriate. We believe that external validation using a two-gate design is not meaningful. If a sequential dataset was available, we validated Ali-M3. 

3. The free access to the commercial tool and online data storage IS a financial support. Please clarify this in the conflict of interest statement.

Reply:

Thank you for your comment. We apologize for the complex COI situation. As previously described, during the study period, Ali-M3 was not a commercial tool. For use in the actual diagnosis of COVID-19, we needed to receive approval immediately. This is due to the worldwide pandemic emergency. In a routine situation, we believe our paper would have been published before Ali-M3 was approved and commercialized. According to your recommendation, we have added this progress to the COI as follows.

Competing interests (COI).

The authors declare no competing interests. The authors did not receive financial funding for this study. At the initiation of the study, Ali-M3 existed as a tool produced by Alibaba Damo (Hangzhou) Technology Co., Ltd for research use that had not received any approval. The results of this study confirm that Ali-M3 has clinical benefits. Therefore, under a special expedited review in Japan, Ali-M3 has been approved by the Japanese Pharmaceuticals and Medical Devices Agency (PMDA) and licensed for use as a diagnostic tool in actual practice. (Approval number: 30200BZX00212000, Datasheet: https://www.pmda.go.jp/files/000235943.pdf) For the license in Japan, Ali-M3 should have been a commercial tool. Therefore, it was not anticipated that M3 Inc. would benefit from the commercialization of the Ali-M3 because of our research. M3 provided the Ali-M3 and storage free of charge. 

4. The current cohort consists of 617 patients, with 289 COVID-19 positive patients, and 223 patients with severe symptoms (needing oxygen support). The practical situation has many more COVID-19 negative patients. Considering the specificity is only 43.2% using the Ali-m3 score threshold 0.2, please clarify how to handle the increasing high number of false positives.

Reply:

Thank you for your important comments.

Your comments depend on the situation using this model. As described in the Discussion, we warned against the use of Ali-M3 as a screening tool. In this case, we set the target population with high prior probability. Moreover, as described in the discussion, we used Ali-M3 for rule-out (i.e., exclusion), not rule-in. In this situation, sensitivity is suitable for an evaluation. Even in serious situations requiring patients to be supplied with oxygen, we must consider rule out as triage. According to your comments, we have added the concern of specificity to limitations.

# Discussion

First, the differentiation performance of Ali-M3 was poor in asymptomatic patients and Ali-M3 did not show good specificity even if the cut-off was changed; thus, Ali-M3 should not be used to screen asymptomatic patients.

5. The results should be strictly discussed. For example, in the Abstract, “sensitivity increased for both cut-off values after 5 days”. But only one threshold 0.2 was mentioned in the Abstract.

Reply:

Thank you for your comment.

We modified the Results section in the abstract as follows.

# Abstract

Results: Of the 617 patients, 289 (46.8%) were RT-PCR-positive. The area under the curve (AUC) of Ali-M3 for predicting a COVID-19 diagnosis was 0.797 (95% confidence interval: 0.762‒0.833) and the goodness-of-fit was P = 0.156. With a cut-off probability of a diagnosis of COVID-19 by Ali-M3 set at 0.5, the sensitivity and specificity were 80.6% and 68.3%, respectively. A cut-off of 0.2 yielded a sensitivity and specificity of 89.2% and 43.2%, respectively. Among the 223 patients who required oxygen, the AUC was 0.825. Sensitivity at a cut-off of 0.5% and 0.2% was 88.7% and 97.9%, respectively. Although the sensitivity was lower when the days from symptom onset were fewer, the sensitivity increased for both cut-off values after 5 days. 

6. And for the “223 patients who required oxygen support”, it’s misleading to skip mentioning the specificity. If we set the threshold to the extreme value (like 0), we can get 100% in sensitivity. But that is not an intelligent tool.

Reply:

Thank you for your comments.

Although we discussed the issue by shifting the threshold, which may be misleading, we only discussed the threshold fixed at 0.2 and 0.5. We did not arbitrarily change these values in the discussion. However,, in clinical practice, it is important to shift threshold values according to the purpose of use. Depending on the clinical situation, the importance of clinical differentials in “misclassification cost” differs. Clinicians are uninterested in performance across all thresholds; they focus on clinically relevant thresholds. Because we could not locate a clinically relevant threshold from all thresholds, we focused on the threshold at 0.2 and 0.5. 

We mentioned the specificity of 223 patients who required oxygen in the Results section. What we want to discuss is that Ali-M3 is suitable for the rule-out of COVID-19. To accomplish this, we required information on sensitivity and not specificity. Therefore, we believe that specificity in the sensitivity analysis is not required in the Abstract. Your concern regarding misleading is important. We modified the conclusion in the Abstract section as follows:

# Abstract

Conclusion: We evaluated the Ali-M3 using external validation. Because Ali-M3 showed sufficient sensitivity performance although lower specificity performance, it was deemed useful in excluding a diagnosis of COVID-19.

7. The commercial provider for Ali-m3 has a website in Japanese only. It’s impossible to review whether this company is a solid AI company or maybe just a contractor of this tool Ali-m3. So the quality and stability of Ali-m3 is unpredictable.

Reply:

Thank you for your comments. 

Unfortunately, there is no specific description for the AI sector on M3's website (https://corporate.m3.com/en/). However, information for Ali-M3 was in the same site (https://corporate.m3.com/en/ir/20200629_2/Microsoft%20Word%20-%20AI_Ali-M3_APPROVAL_E.pdf). The quality and stability of Ali-M3 were officially confirmed by the Japanese PMDA using our data on June 29, 2020. (https://www.pmda.go.jp/english/about-pmda/0002.html).

# Acknowledgement

Ali-M3 was officially approved by the Japanese PMDA using our data on June 29, 2020. (https://www.pmda.go.jp/english/about-pmda/0002.html).

8. Does Ali-m3 have a medical license approved by some governmental agencies?

Reply: 

Yes. PMDA approved Ali-M3 (# 30200BZX00212000) on June 29, 2020. We have added this fact to Acknowledgement as follows:

Ali-M3 was officially approved by the Japanese PMDA using our data on June 29, 2020. (https://www.pmda.go.jp/english/about-pmda/0002.html).

9. This study cited the commercial tool Ali-m3 by an internal report of a commercial company, which is not the service provider “m3”. Please clarify this.

Reply: 

Thank you for your comments. As previously described, at the initiation of the study, M3 was only one of the channels for Japanese researchers to connect with Alibaba. Ali-M3 was developed by Alibaba Damo Technology Co., Ltd. During this time, Ali-M3 did not have a name. The name Ali-M3 was provided in the process of approval. The Ali-M3 datasheet was provided by Alibaba Damo Technology Co., Ltd.

10. And what is the online like to the validated tool Ali-m3? It’s not acceptable to ask the anonymous reviewer to contact the commercial provider to access the cloud-based tool.

Reply:

Thank you for your comments.

As mentioned earlier, Ali-M3 was approved by PMDA. Approval letters are available on the web, although they are in Japanese. (https://www.pmda.go.jp/files/000235943.pdf) We translated the instructions as follows. 

1. Preparation for use:

(1) Turn on the general-purpose IT equipment to be installed or access the product on the cloud server.

(2) Start the product.

2. Operation:

(1) Input the X-ray CT image from the X-ray CT diagnostic equipment or the server that stores these images.

(2) The confidence level of the CT image findings in COVID-19 pneumonia is presented, and the area of interest is marked on the image.

(3) Save the results.

3. Exit.

(1) Click on the exit icon on the screen or select the menu items' exit function to exit the product.

(2) If necessary, turn off the general IT equipment.

The screen image of CT is like the one below.

Reviewer #2: The manuscript is about a system for real-time sentiment prediction on Twitter streaming data for coronavirus pandemic. The paper is well-organised, but I still have some concerns:

Reply:

Thank you for your constructive comments.

However, we did not discuss “a system for real-time sentiment prediction” on Twitter streaming data for the coronavirus pandemic. 

Reviewer #3: The contribution of this research paper isn't clear. Sorry to say that, however, I can't get the point of this paper from the manuscript. Although you state your purpose as "Ali-M3, ... However, Ali-M3 has not been externally validated.", this statement didn't show anything about what you want to do in this research paper.

Reply:

Thank you for your clear comments. 

Our aim was to perform external validation of the Ali-M3. Similar to other AI systems for diagnosing COVID-19, Ali-M3 has high accuracy in the process of internal validation. In usual prediction models, after developing a prediction model, it is strongly recommended to evaluate the model's performance with other participant data than was used for model’s development. Moreover, AI makes the model’s data fit better than existing statistical methods. Therefore, AI can easily cause overfitting. Thus, external validation is more important than the usual model development in an AI model’s development. Although limited, we evaluated the actual specifications of Ali-M3 using external validation with a later period, different countries, and different setting datasets. Our dataset is one of the ideal datasets from the perspective of the TRIPOD statement. According to your comments, we have modified the abstract as follows:

# Abstract

Background: Ali-M3, an artificial intelligence program, analyzes chest computed tomography (CT) and detects the likelihood of coronavirus disease (COVID-19) in a range of 0 to 1. However, Ali-M3 has not been externally validated. Our aim was to evaluate the accuracy of Ali-M3 in identifying COVID-19 and to discuss its’ clinical value.

Purpose: To evaluate the external validity of the Ali-M3 using sequential Japanese sampling data.

Ref: Moons KG, Altman DG, Reitsma JB, et al. Transparent reporting of a multivariable prediction model for individual prognosis or diagnosis (TRIPOD): explanation and elaboration. Ann Intern Med. 2015; 162:W1-73. [PMID: 25560730] doi:10.7326/M14-0698

Based on the conclusion of this paper, "Our results indicated that AI-based CT diagnosis could be useful for ...", it seems that you want to prove that Ali-M3 can be used to diagnose COVID-19, but the data samples used to evaluate Ali-M3 and the results are not good enough to support your conclusion. There are only several hundreds of samples in your evaluation process, even more, you didn't provide background information about those samples, such as how were they collected and which groups of people they covered. So, in my opinion, they can't represent all COVID-19 situation.

Reply:

Thank you for your comments regarding the sample size and generalizability.

* Sample Size

In external validation of the prediction model using dichotomous outcomes, the number of both events and non-events in the external validation cohorts is recommended to be over 200, at least in each cohort. (Vergouwe Y, 2005 Journal of Clinical Epidemiology) In this cohort, 289 patients were RT-PCR (+), and 460 patients were RT-PCR (-). Thus, because our cohort sample was sufficient for the external validation study, your suggestion, which is short of sample size, was not suitable for our cohort.

* Generalizability

We also believe that this external validation cannot be adapted for all situations in which physicians suspect COVID-19. The limitations of the adaptation of this external validation study are necessary. In this external validation, the facility was limited to 11 Japanese tertiary care facilities that provided treatment for COVID-19 in each region of the country. These are areas where facilities are COVID-19 hot spots in Japan. During the study period, patients suspected of COVID-19 were integrated into these facilities in each region, regardless of their severity. Whether or not and when patients should be seen for suspected COVID-19 was controlled by the health authorities. Patients were instructed to go to the healthcare facility that they were integrated into if they had symptoms of suspected infection. These included persistent fever or cough, or if they had been in close contact with someone already known to be infected. Physicians decided whether patients required COVID-19 testing. Although the threshold that physicians used for suspected COVID-19 must have been different for each physician (because there was a shortage of information at the initiation of this study), almost all patients suspected of COVID-19 received RT-PCR and chest CT in each facility. This was because physicians, who diagnosed patients with suspected COVID-19, were aware of the information that COVID-19 patients might have some features on chest CT. The records of RT-PCR were not allowed to have any missing data due to Japanese domestic laws regarding COVID-19. Sampling was performed sequentially. Therefore, potentially eligible participants were identified on physicians' advice when patients from Japanese COVID-19 hot spots presented with symptoms and were suspected of having COVID-19. Moreover, patients who were included required both RT-PCR tests and chest CT.

Because we agreed with your opinion that our cohort could not represent all COVID-19 situations, we modified the Conclusion section in the Abstract and the Body of the manuscript.

# Abstract

Conclusion: We evaluated Ali-M3 by external validation using symptomatic patient data from Japanese tertiary care facilities. Because Ali-M3 showed sufficient sensitivity performance although lower specificity performance, Ali-M3 was shown to be useful in excluding a diagnosis of COVID-19.

# Body

We conducted a retrospective cohort study for external validation of Ali-M3 using symptomatic patient data from Japanese tertiary care facilities. Despite limited data analysis, our results indicated that AI-based CT diagnosis could be useful for a diagnosis of the exclusion of COVID-19 in symptomatic patients. This is particularly true in patients requiring oxygen and with only a few days after symptom onset.

Reference:

Vergouwe Y, Steyerberg EW, Eijkemans MJC, Habbema JDF (2005) Substantial effective sample sizes were required for external validation studies of predictive logistic regression models. J Clin Epidemiol 58:475–483

Besides the insufficient testing samples, the performance of the model with AUC 0.79, 0.82 isn't very good. How could a model with such performance be used in COVID-19 diagnosis?

Reply: 

Thank you for your comments regarding model performance and use. 

As you mentioned, we used AUC, sensitivity, and specificity. It is well established that diagnostic tests are best understood when presented in terms of gains and losses to individual patients [11]. The AUC lacks clinical interpretability because it does not reflect this fact. Clinicians are not interested in performance across all thresholds that AUC provides. They focus on clinically relevant thresholds. Thus, evaluation using AUC only will not produce relevant information for clinical practice. To our knowledge, the validation of the CT diagnosis system for COVID-19 using a sequential dataset (consecutive sampling) did not show an excellent AUC in the validation set. (Ref)

We discuss the usefulness of Ali-M3 in the diagnosis of exclusion of COVID-19. As noted in the discussion, RT-PCR is not a perfect test. Even now rapid tests are becoming available. Testing accuracy depends on the viral load and the instability of the collection method. Furthermore, there is a concern regarding the emergence of variants of COVID-19 that bypass the currently used RT-PCR, as reported in Brittany. To cope with this situation, it is important to use multiple diagnostic modalities. We strongly believe that the diagnosis of exclusion by Ali-M3 has clinical implications.

Another question, what is your work in this research? From the manuscript, I see that you ran the Ali-M3 model which is already a usable deep learning model, with patients data which I don't know you collected it or not, and take some simple analysis about the results. Are these all you had did in this research? What's the significance of what you did? Maybe you could add more contents in your manuscript about what you did, such as data collection, sample pre-processing, model adjustment, deep analysis, diagnosis direction, practice guideline, or some other things.

Reply:

Thank you for your comment regarding the author contributions. We understand the reviewer's concerns. We have added an explanation of the author contributions as follows:

# Author Contribution

YK, YM, JM, JK, KT, HF, TH, AK, MS, FH, and SI compiled medical and imaging information at each institution. Group author members collected substantive data at each site and added items to the survey that were required for clinical information. TI, YK, and SF were involved in the study design and data interpretation. TI was involved in the data analysis and sample pre-processing. The analysis of the CT by Ali-M3 was carried out by Nobori on behalf of M3. (M3 and Nobori were blinded to the patients' data including the result of RT-PCR) All authors critically revised the report, commented on drafts of the manuscript, and approved the final report. 

A lot of analysis were done focusing on cut-off point adjustment. However what's the meaning of those analysis? Sensitivity and specificity have big changes when you use different cut-off values and they can be affected by the ratio of positive and negative samples of testing dataset. So I think it's not necessary to analysis those values because they can't represent real performance of prediction model.

Reply:

Thank you for your comments.

Again, it is the sensitivity and specificity, not the AUC, which are relevant in actual clinical practice. It is the sensitivity and specificity of the test as a function of a certain threshold value that is useful for an actual diagnosis. For this reason, we developed a number of threshold values and discussed the sensitivity and specificity of these values. We believe that it is the diagnosis of exclusion of COVID-19 that is more important when considering the use of the test. We believe that it is important for clinicians to have both a threshold value of 0.2, and a threshold value of 0.5, which is close to the Youden-index threshold.

---

## [Decision Letter · Decision Letter 1]

6 Oct 2021

Accuracy of deep learning-based computed tomography diagnostic system of COVID-19: A consecutive sampling external validation cohort study

PONE-D-21-03621R1

Dear Dr. Ikenoue,

We’re pleased to inform you that your manuscript has been judged scientifically suitable for publication and will be formally accepted for publication once it meets all outstanding technical requirements.

Kind regards,

Haoran Xie

Academic Editor

PLOS ONE

Additional Editor Comments (optional):

Reviewers' comments:

Reviewer's Responses to Questions

**Comments to the Author**

1. If the authors have adequately addressed your comments raised in a previous round of review and you feel that this manuscript is now acceptable for publication, you may indicate that here to bypass the “Comments to the Author” section, enter your conflict of interest statement in the “Confidential to Editor” section, and submit your "Accept" recommendation.

Reviewer #2: (No Response)

Reviewer #3: All comments have been addressed

2. Is the manuscript technically sound, and do the data support the conclusions?

Reviewer #2: Yes

Reviewer #3: Yes

3. Has the statistical analysis been performed appropriately and rigorously? 

Reviewer #2: Yes

Reviewer #3: Yes

4. Have the authors made all data underlying the findings in their manuscript fully available?

Reviewer #2: No

Reviewer #3: (No Response)

5. Is the manuscript presented in an intelligible fashion and written in standard English?

Reviewer #2: Yes

Reviewer #3: (No Response)

6. Review Comments to the Author

Reviewer #2: In my idea, although this study has a lot of limitations, it can be a good start to use AI in clinics.

Reviewer #3: (No Response)

7. PLOS authors have the option to publish the peer review history of their article (what does this mean?). If published, this will include your full peer review and any attached files.

Reviewer #2: No

Reviewer #3: No

---

## [Editor Report · Acceptance letter]

19 Oct 2021

PONE-D-21-03621R1 

Accuracy of deep learning-based computed tomography diagnostic system for COVID-19: A consecutive sampling external validation cohort study 

Dear Dr. Ikenoue:

I'm pleased to inform you that your manuscript has been deemed suitable for publication in PLOS ONE. Congratulations! Your manuscript is now with our production department. 

Kind regards, 

on behalf of

Professor Haoran Xie 

Academic Editor

PLOS ONE